# Haplotype Analysis of *BADH1* by Next-Generation Sequencing Reveals Association with Salt Tolerance in Rice during Domestication

**DOI:** 10.3390/ijms22147578

**Published:** 2021-07-15

**Authors:** Myeong-Hyeon Min, Thant Zin Maung, Yuan Cao, Rungnapa Phitaktansakul, Gang-Seob Lee, Sang-Ho Chu, Kyu-Won Kim, Yong-Jin Park

**Affiliations:** 1Department of Plant Resources, College of Industrial Science, Kongju National University, Yesan 32439, Korea; mmh7272@gmail.com (M.-H.M.); tzmaung.yau2009@gmail.com (T.Z.M.); yuancao2017@gmail.com (Y.C.); phitaktansakul2017@gmail.com (R.P.); 2Agricultural Biotechnology Department, National Institute of Agricultural Sciences, Rural Development Administration, Jeonju 54874, Korea; kangslee@korea.kr (G.-S.L.); kyuwonkim@kongju.ac.kr (K.-W.K.); 3Center of Crop Breeding on Omics and Artificial Intelligence, Kongju National University, Yesan 32439, Korea; sanghochu76@gmail.com

**Keywords:** betaine aldehyde dehydrogenase 1 (BADH1), salt stress, domestication, cultivated rice, wild rice

## Abstract

Betaine aldehyde dehydrogenase 1 (*BADH1)*, a paralog of the fragrance gene *BADH2*, is known to be associated with salt stress through the accumulation of synthesized glycine betaine (GB), which is involved in the response to abiotic stresses. Despite the unclear association between *BADH1* and salt stress, we observed the responses of eight phenotypic characteristics (germination percentage (GP), germination energy (GE), germination index (GI), mean germination time (MGT), germination rate (GR), shoot length (SL), root length (RL), and total dry weight (TDW)) to salt stress during the germination stage of 475 rice accessions to investigate their association with *BADH1* haplotypes. We found a total of 116 SNPs and 77 InDels in the whole *BADH1* gene region, representing 39 haplotypes. Twenty-nine haplotypes representing 27 mutated alleles (two InDels and 25 SNPs) were highly (*p* < 0.05) associated with salt stress, including the five SNPs that have been previously reported to be associated with salt tolerance. We observed three predominant haplotypes associated with salt tolerance, Hap_2, Hap_18, and Hap_23, which were Indica specific, indicating a comparatively high number of rice accessions among the associated haplotypes. Eight plant parameters (phenotypes) also showed clear responses to salt stress, and except for MGT (mean germination time), all were positively correlated with each other. Different signatures of domestication for *BADH1* were detected in cultivated rice by identifying the highest and lowest Tajima’s D values of two major cultivated ecotypes (Temperate Japonica and Indica). Our findings on these significant associations and *BADH1* evolution to plant traits can be useful for future research development related to its gene expression.

## 1. Introduction 

Soil salinity has become important as one of the major constraints affecting rice production worldwide, and accordingly, breeding approaches using marker-assisted selection or genetic engineering to produce salt-tolerant varieties are also being developed [1]. Soil salinization is a serious problem in rice cultivation [2], especially at the germination stage.

A homologous gene of betaine aldehyde dehydrogenase 2, also known as *BADH1*, was reported to be a candidate gene with a close correlation with salt tolerance [3]. Glycine betaine (GB) has been reported as an osmo-protectant compound [1] that can be synthesized in many living organisms, including plants and animals, in response to abiotic stresses, such as salt, drought, and temperature [4]. Glycine betaine can protect protein structure and enzyme activity and stabilize membranes to establish osmotic and ionic stress in plants [3]. GB is also widely distributed in bacteria, algae, and higher plants such as sugar beet and cotton [3,5]. In higher plants, GB can be synthesized by the two-step oxidation of choline by ferritin-dependent choline monooxygenase (CMO) and betaine aldehyde dehydrogenase (*BADH*). The enzyme betaine aldehyde dehydrogenase (*BADH*) has been reported to be responsible for GB synthesis, and many plant species have been identified as potential GB accumulators [4].

However, although some reports on *BADH1* suggested its association with salt tolerance, the function of *BADH1* is unclear, and there are even conflicting reports from various studies on the relationship between *BADH1* and salt stress tolerance in rice. The possibility has been considered that GB cannot accumulate in rice due to the lack of a functional CMO gene [6]. Protein modified by SNP substitutions of the *BADH1* gene indicated its specific association with aroma rather than salt tolerance [7]. Despite the unclear and conflicting reports on the relationship between *BADH1* and salt stress tolerance in rice, there are some indirect findings regarding the relationship. The gene expression of *BADH1* in rice indicates that the gene encodes a key enzyme for GB biosynthesis and is closely related to salt tolerance [8]. Increased levels of *BADH1* transcripts in salt-stressed Japonica and Indica nonfragrant rice varieties [9,10] suggest the possible association of the *BADH1* gene with salt tolerance through an undetermined mechanism.

Recent developments in sequencing technology have simplified the analyses of single nucleotide polymorphisms (SNPs) and insertions and deletions (InDels), which are the basis for allele differentiation. Haplotype analysis, which includes a set of linked SNPs, is more informative than the analysis of a single SNP in determining the associations with phenotypes [7]. Extensive research on the *BADH1* gene has not yet been conducted at the genetic and molecular levels [11], and only seven SNPs in certain exons (Appendix A) have been reported [12].

Here, to confirm the association between *BADH1* and salt tolerance, we used sophisticated sequencing technology (1) to investigate the genetic diversity of *BADH1*, (2) to examine the haplotype variation within the gene region of *BADH1*, and (3) to observe the association between the main haplotypes of *BADH1* and salt tolerance at the germination stage of tested rice accessions. We also conducted population genetic studies, such as nucleotide diversity, population structure, and Tajima’s D, and phylogenetic studies.

## 2. Results

### 2.1. Discovery of Genetic Variations in BADH1

To investigate how many variations and types occurred, we conducted variant calling of 475 rice accessions and extracted all the variants within the gene region of *BADHI* by using VCFtools. The results revealed four different types of variants, single nucleotide polymorphism (SNP), insertion (Ins), deletion (Del), and structure variation (SV), and we identified variant numbers for the classified subpopulations of cultivated rice and wild rice (Table 1). According to the summarized number, we observed that SNPs represented the highest number of variants for all classified subpopulations, among which the wild showed the highest number. Among the cultivated subpopulations, Indica and Aus had the same number of SNPs (23), representing the highest value except when compared with the wild rice (105). The wild rice group showed a higher number of identified variants, 38 insertions, 36 deletions, and 2 structural variations (the detailed observed number of variants for each wild rice accession is provided in Appendix A). Overall, we noticed that the wild rice had a higher number of different variants than any of the cultivated subpopulations.

### 2.2. Population Structure, Principal Component Analysis (PCA), and Fixation Index (F_ST_ Test) of BADH1

To verify the subgroups (wild and cultivated subgroups) followed by genotypic variants, we conducted a Bayesian Analysis of Population Structure (BAPS) version 6.0 and PCA. The structure was implemented by increasing K values from 2 to 7 (Figure 1A). Except for K = 2, the cultivated rice was clearly separated from the wild, but their subpopulations were mixed in K values of 3 and 4. Temperate Japonica and Tropical Japonica were mixing always in all K values. At K = 3 and 4, Indica was also admixed with the Japonica group, showing its mixed structure with other minor cultivated subgroups (Aus, Aromatic, and Admixture). All the cultivated subpopulations and wild subpopulation were clearly separated from each other at K values 5, 6, and 7, but the internal subgroups of the wild variants were mixing while the others’ internal subgroups were separated. Overall, the wild had internal subgroups at every K value in spite of their clear separation from cultivated subgroups.

We conducted a PCA analysis to multivariate the original variant datasets of *BADH1* into a dimensional scaling display (Figure 1B) and observed the associations between the classified ecotypes. According to the display, the wild, Indica, and Temperate Japonica should have been grouped separately but in actuality were not at all. Some Indica were mixed with Tropical Japonica, Admixture and Aus while Temperate Japonica were mixed with Aromatic, some wild, and Indica. Overall, within the cultivated subpopulations, the associations were greatly distant compared with their distance from wild rice.

To make sure the population analysis and further finding, we used a sophisticated method, *F_ST_* value (fixation index), which indicates the genetic distance or differentiation between the populations or subpopulations. We calculated *F_ST_* values for the classified ecotypes of cultivated rice and checked their distances from the wild within the *BADHI* region. The highest *F_ST_* value (0.6786) was found between Temperate Japonica and Tropical Japonica, followed by the pair, Temperate Japonica–Indica (0.6062), while the lowest was represented between Tropical Japonica and the wild (0.0376) (Figure 1C). *F_ST_* values between the cultivated subpopulations (both Japonica, Indica) are higher than those between cultivated and wild rice.

### 2.3. Genetic Diversity of BADH1

We calculated the nucleotide diversity value of *BADHI* in 475 rice accessions to investigate the degree of polymorphisms at different segregating sites by means of comparing different genotype sequences. The diversity values were analyzed based on the classified subpopulations/groups to be compared. We also calculated diversity values for the whole rice collection and for major subgroups/ecotypes, namely, Temperate Japonica, Tropical Japonica, and Indica. We omitted other cultivated minor subgroups, aus, aroma, and admixture, due to the very low number of rice accessions. To observe the clear diversity level, we compared the resulting diversity values (π) of classified cultivated ecotypes to those of wild as well as whole accessions. We found that Indica showed its highest diversity value (Figure 2A) across the *BADHI* gene region, while both Japonicas showed the lowest nucleotide diversity. The nucleotide diversity of wild group was between that of the two higher-diversity groups and the two lower-diversity groups. As shown in (Appendix A), Indica (0.00435) was much higher in value than the lowest Temperate Japonica (0.00005). The nucleotide diversity of Indica and wild rice were similar each other.

To investigate the differences between the observed nucleotide diversity and expected nucleotide diversity due to selection, we calculated Tajima’s D values, which were determined by a pairwise comparison and their segregation sites number for the same classified groups. The calculated values ranged from the lowest, −1.4551 (Temperate Japonica) to the highest, 3.1529 (Indica) (Figure 2B and Appendix A). The signatures of two directional selective sweeps were observed for two major ecotypes of cultivated subpopulations. The Tajima’s D over *BADH1* for Indica, one of major cultivated subpopulations, was particularly higher than the others, indicating that *BADH1* of Indica had undergone balancing selection; whereas Temperate Japonica, another major type of cultivated rice, showed the lowest value, indicating that *BADH1* of Temperate Japonica had undergone purifying selection. Although the nucleotide diversity of Indica and wild rice is similar, a balancing selection was observed only in Indica.

### 2.4. Phylogenetic Study of BADH1

We constructed a phylogenetic tree to observe the evolutionary relationships of *BADHI* in 475 rice accessions by their genotypic differences or similarities in terms of ecotypes/subpopulations (Figure 3). This analysis was conducted mainly to observe the evolutionary studies of cultivated rice accessions and their relatedness to different wild accessions based on ecotype. The cultivated subgroups were dispersed across different separated tree branches of wild species. Indica (33.3%) was associated with *O. nivara* as well as with some aus in the same clade. Temperate Japonica was rooted separately and directly from common ancestors. Most of the wild rice was genetically distant from cultivated ecotypes, especially *O. meridionalis*, *O. punctata*, and *O. longistaminata*.

### 2.5. Haplotype Diversity

To identify the association between *BADH1* and salt tolerance at the haplotype level, we conducted haplotype diversity analysis on the whole genomes of 475 rice accessions. We analyzed haplotype diversity only on 421 types of cultivated rice (Appendix A) first and then compared it to that of 54 types of wild rice (Appendix A). There were 116 SNPs and 77 InDels covering all the identified exons and introns within the *BADHI* gene region.

In cultivated rice, we verified 39 haplotypes (hereafter referred to as “Hap”) representing genetically identified variants, but in Figure 4, we show only functional SNPs (fSNPs) observed in exons (maf < 0.03). Then, we also classified specific rice ecotype (Indica, Temperate Japonica, and Tropical Japonica) for all the rice accessions under each haplotype, and their respective accession numbers were also provided together with their haplotype number. Based on those total identified rice accession under each haplotype, we considered five major haplotypes indicated in the largest numbers of rice accessions, Hap_2, Hap_3, Hap_4, Hap_18, and Hap_23, by more than five accessions. The total number of rice accessions separately represented by each subpopulation were also listed by haplotype; then, we specifically focused on two such two haplotypes, Hap_18 and Hap_23, for their associated functional alleles (mutation sites). Hap_3 was the first major haplotype (62.7% of all cultivated rice) by which all the rice accessions showed the same sequence as the reference. Hap_18 (40 Indica) and Hap_23 (14 Indica) belonged only to Indica, showing the same fSNPs, G/T in exon 11 and A/T in exon 4. Referring to those haplotypes, we investigated wild haplotypes (in this case, haplotyping of wild rice) to examine the genetic variation in *BADHI,* and we found six wild haplotypes showing the same SNP (G/T) as Hap_18, only four of which showed the same SNP (A/T) as Hap_23 (Appendix A). In the wild, we found many functional SNPs that addressed almost all 50 verified haplotypes, but only six haplotypes had the same SNP substitutions as those of cultivated haplotypes (Indica). When the cultivated and wild haplotypes were compared in terms of InDel variants, we found no InDel variants in exon regions, but in the intron regions, we found two InDel variants (Appendix A).

To determine the genetic association of the *BADHI* gene among the classified subpopulations of cultivated rice and wild rice, we constructed a network using previously identified haplotypes. In this case, we investigated the association of five major cultivated haplotypes (due to their highest number of rice accessions they occupied) with wild rice accessions. We generated 50 haplotypes referring to 54 accessions of 21 different species. Using all-wild haplotypes and selected cultivated haplotypes, we constructed TCS network in the PopART program (Figure 5). Hap_3 and Hap_4 were grouped in the same clade. Two cultivated haplotypes, Hap_2 and Hap_3,4 (combined haplotype for Hap_3 and Hap_4), were closely related by the smallest number of mutational steps. Only one wild haplotype, belonging to *Oryza australiensis−1*, was related to cultivated Hap_3,4 at a closer distance than all others. Interestingly, two cultivated haplotypes, Hap_18 and Hap_23, both belonging only to Indica, were also distantly associated with wild haplotypes and even members of the same group of cultivated haplotypes, Hap_2 and Hap_3,4. All the identified wild haplotypes were genetically far from each other in the gene region of *BADH1*. This relatedness of wild haplotypes indirectly agreed with the discovery of the highest number of different variants (Table 1 and Appendix A).

### 2.6. Screening and Evaluation of Salt Tolerance Phenotypes

*BADHI* is an important synthetic enzyme involved in the response mechanism to environmental stresses, especially salt tolerance. To perform a deep study of the responses of this gene *BADHI* to plant characteristics, we screened eight major plant parameters, germination percentage (GP), germination energy (GE), germination index (GI), mean germination time (MGT), germination rate (GR), shoot length (SL), root length (RL), and total dry weight (TDW), of 417 cultivated rice plants under a range of salt stress conditions (200 mM NaCl) together with the corresponding control (0 mM NaCl) during their germination stage (Appendix A). Descriptive statistics were first checked among the calculated mean values of major traits under both conditions (Table 2). Then, pairwise correlations were also checked by Pearson coefficients among the major traits (Appendix A). The statistical analysis revealed that all the tested phenotypes (parameters), except MGT and TDW, indicated their responses (in terms of lower mean values) to salt treatment compared with their respective values under the control condition. Data ranges were varied based on the trait types, and all these phenotypes (traits) during seed germination were obviously and negatively influenced by salt treatment. The response of each trait to salt treatment was consistent with our previous finding where rice seedlings were negatively affected regarding shoot length (SL), root length (RL), and total dry weight (TDW) by salinity stress [13]. In the case of pairwise comparison by their correlation coefficients under the control treatment, traits such as GP, GE, GI, and GP were positively and significantly correlated with each other, while all four parameters were negatively and significantly correlated with MGT.

### 2.7. Test/Control Ratio of Eight Major Plant Parameters

The test/control ratio (the ratio of the measured value in the test condition to the measured value in the control) in the salt tolerance experiment of each trait was calculated from the ratio of recorded values under the test condition (200 mM NaCl) to those of the control (0 mM NaCl). We screened the calculated relative values of all eight phenotypic parameters and analyzed their significant differences among ecotypes (Figure 6A–H) and Appendix A). The analyzed values revealed that almost all plant parameters had the same trend of response to salt treatment among the ecotypes. For example, there were no significant differences in the relative values of GE, GI, GR, and MGT (Figure 6B–E), but in TDW, Temperate Japonica had a significant difference from with the other ecotypes (Figure 6H). In the case of RL and SL, Japonica ecotypes had higher traits in response to salt treatment (Figure 6F,G), and again in GP, Japonica plants were significantly different from Aromatic, Aus, and Indica (Figure 6A). Although ecotype groups did not differ from those of each other in some traits, if they were observed to be different, Japonica ecotypes were mostly significantly or comparatively differentiated from others.

### 2.8. Association of BADH1 Haplotypes and Plant Parameters under Salt Stress

Haplotype analysis of all cultivated rice is presented above. Here, we analyzed the association between the phenotype data and the gene region of *BADH1* in a general linear model (GLM) by the TASSEL 5 software program. The resulting marker positions were selected based on higher *p*-values. After identifying markers at the *p*-value (<5%) within the gene region of *BADH1*, we found 27 marker positions correlating to recorded plant major traits, and several traits belonged to one marker position (Appendix A). Those identified marker positions were checked and merged with the previously extracted variants (SNPs/InDels) represented by 39 haplotypes.

In the case of haplotypes that met such identified marker positions for eight major phenotypes, we found that 10 of 39 haplotypes did not have any mutated variants, so only 29 haplotypes were represented by the identified marker positions for major traits (Appendix A). Among those 29 haplotypes correlated to the positions identified in the analysis of salt-tolerant phenotypic traits, we observed that there were five major haplotypes, indicating that their highest rice accession numbers were represented by Temperate Japonica, Indica, and Tropical Japonica. We analyzed the associations of these five major haplotypes (Hap_2, Hap_3, Hap_4, Hap_18, and Hap_23) with each of eight previously identified plant parameters (Figure 7A–H and Appendix A). We used Scheffe’s test to indicate the significant difference among the comparison of those haplotypes for each parameter at *p*-values (<0.05). For the plant parameters rGP and rGI, we identified Hap_18 as a significant group, which was significantly affected by salt treatment (Figure 7A,C). Figure 7D,E, representing rMGT and rGR, showed nonsignificant differences in plant responses to salt treatment. Hap_4 (Figure 7F) was significantly different from the others in rRL analysis. For other plant parameters (Figure 7B,G,H), no highly significant responses were found among haplotypes, but Hap_4 was highly significantly different from Hap_2 and Hap_18 in rGE and was highly significantly different from Hap_2, Hap_18, and Hap_23 in rSL analysis. Hap_2 was also highly significantly different from Hap_18 in rTDW. Overall, all the tested plant parameters were relatively influenced by salt stress at the germination stage. Interestingly, among the selected haplotypes to be analyzed for association testing, two haplotypes, Hap_18 and Hap_23, belonged to only Indica rice accessions. Then, according to rGP and rGI, a significant effect of salt was observed in Hap_18.

## 3. Discussion

Whether *BADH1* is mainly associated with aroma or salt stress in rice is unclear, since no association was found between *BADH1* haplotypes and salt tolerance [7], but aromatic rice has a specific association with *BADH1*. A recent paper again indicated that the *BADH1* transcript level was comparatively increased during salt treatment [14] and that it, as a homolog of the *BADH2* gene, can also be induced by environmental factors, such as salt [15]. We also investigated the association of these two orthologous genes, *BADH1* and *BADH2*, by tracing back their ancestral histories in 19 rice species (Appendix A). According to phylogenetic display, we noticed that both genes were localized in Japonica rice species but in different clades. As a result of their evolution from a common ancestor, the two *BADH* enzymes are high in sequence homology. However, their transcriptional responses to salt would still be implicated by upregulated expression of *BADH1* to salt and drought stress [16] as well as their specific association with aroma by protein modeling [7]. The inconsistent findings could be attributable to differences either in rice germplasm materials or the growth stages investigated in their studies.

We used whole genome data of 475 Korean rice accessions collected worldwide to investigate the genetic diversity and domestication information on *BADH1* and plant parameters of salt tolerance with an association study. In this case, we picked up only the gene region of *BADH1* (23171516–23176332), in another way, chromosome 4; then, possible genetic variations were discovered by using VCFtools. Variant calling on *BADH1* revealed 116 SNPs, 38 Ins, and 39 Dels, indicating that a higher number of variants were found in wild rice than in cultivated rice. This may be because compared to Asian rice, wild rice has a complex domestication history that was only recently reconstructed [12,17,18]. Haplotype analysis revealed 39 haplotypes covering 116 SNPs and 77 InDels in both exons and introns, including the untranslated region (UTR). Only 27 genetic variants (SNPs and InDels) represented the identified marker positions analyzed by a generalized linear model (GLM) for the association of salt tolerance traits and the *BADH1* gene region (Appendix A). Only five SNPs that were localized in exons and all five SNPs have been introduced by previous reports (Appendix A). Three SNPs by our previous study were represented by substitutions such as G/C in exon 1, A/G in exon 6, and C/A in exon 12 [19], while two other SNP substitutions, A/T (exon 4) and G/T (exon 11), were discovered by Singh et al. [7]. There remained 22 (two indels and 20 SNPs) genetic variants in introns, which represented 29 haplotypes overall. In terms of ecotypes for those haplotypes, Indica showed the highest accession number, representing three haplotypes (Hap_2, 18, and 23) with 14 new variants (one InDel and 13 SNPs) in introns. Thailand et al. have reported that the expression of the *BADH1* gene in Indica correlates with salt stress and other environmental stresses, such as plasmolysis, temperature, and light [15]. Rice and its major trait domestication have been independent of two different species, African rice (*Oryza glaberrima* Steud) and Asian rice (*Oryza sativa* L.) [20], although there could be a third domestication event by recent archeological evidence in the Amazon [21]. Genomics has produced unprecedented amounts of datasets for deeper insights into domestication studies [20], and whole-genome resequencing of rice DNA could result in tracing back the details of its domestication history by population genomic analysis [22]. Resequencing our previous whole-genome data also revealed different domestication patterns of *BADH1* and *BADH2* [19].

Here, we could find that the clue of our result of the association between genotype/haplotype and salt tolerance was due to the process of adaptation for human use rather than a random process based on two independent analytic results, population structure and selective sweep. As expected, clear separations of classified cultivated subpopulations from wild were observed at most of the K values, especially at 5, 6, and 7, indicating the existence of cultivated ecotypes. The cultivated ecotypes especially between Japonica and Indica were clearly separated from each other by PC1 and PC2 in PCA, and a following analysis of population differentiation via *F_ST_* showed a considerably high *F_ST_* value between Japonica and Indica, indicating genetic isolation from each other. The general phenomenon of domestication in plants or animals is the reduction of genetic diversity via genetic erosion [23], and different domestication pathways of rice genes have recently been updated through modifications of morphological traits, physiological characteristics, and ecological adaptability from the wild into modern cultivated rice [24]. In our study, we found a relatively high genetic diversity in Indica compared to Japonica rice groups. This may be due to the increase of heterozygotes by human-mediated hybridization during cultivation or breeding programs of Indica rice accessions. There was an interesting finding we noticed: most of the haplotypes were Indica rice accessions that had genetic markers (alleles) associated to salt-tolerant plant parameters. These findings also seemed to be clues of highly diverse speciation of Indica rice. Tajima’s D, which signifies selective sweep by observed frequency polymorphisms relative to expectation, showed the lowest value for Temperate Japonica and the highest value for Indica, suggesting the opposite directional signification between Temperate Japonica (purifying selection) and Indica (balancing selection). The above two results of opposite directional signification suggest that the association between *BADH1* and salt tolerance may be the result of the independent domestication of Japonica and Indica involving their genetic isolation. Similar findings but opposite domestication signatures for Japonica (Tajima’s D = 4.47) and Indica (Tajima’s D = −1.33) were reported by GWAS analysis [25]; one potential candidate gene *OsSTL1* (salt tolerance level) identified on chromosome 4 was higher in allele frequency in Indica than Japonica, improving the overall salt tolerance. However, our previous results reported an inconsistent finding: the lack of domestication in Asian rice might be due to the small amount of difference detected in the signature of selective sweep for Japonica (Figure 1B) [19].

Germination is considered to be one of the most critical steps in the life cycle of a crop. Due to salinity problems at the germination stage [2], the breeding of salt-tolerant varieties, especially during germination, has been increasing in agricultural countries around the globe. Many research experiments on the effects of environmental conditions have been reported by many research groups, particularly the effects of salinity, drought, cold, heat, light intensity, and CO_2_ at the molecular level [15]. In this study, we found clear physical responses of salt-tolerant plant parameters to 200 mM NaCl compared to the control (0 mM NaCl). All the phenotypes showed clear responses to salt treatment, and except for MGT, all phenotypes were positively correlated with each other in salt-treated conditions (Table 3). In particular, the differences in the recorded values of phenotypes under the control and salinity conditions (0 mM and 200 mM NaCl) indicated that rice growth during the germination stage was significantly inhibited by salt stress, which resulted in a very low germination energy and index (GE and GI), as well as root length (RL) and shoot length (SL), which in turn reduced plant density and even yield. Therefore, the development of salt-tolerant rice germplasms would be indispensable to maintain good plant growth beginning in the early stages of rice cultivation.

All the plant parameters showed responses to salt stress in all selected haplotypes (Figure 7). *BADH* has been playing several important functions in plants, and besides them, it is also considered as an associated gene for many types of abiotic stress tolerance, including drought, osmolarity, submergence, temperature, chilling, ultraviolet radiation, and so on [26]. Among different environmental stresses, the primary response to *BADH1* expression in Indica rice was induced by salt treatment within 24 h [7]. The effects of salt treatment on the germination percentage (GP) and germination index (GI) also highlighted the Indica haplotype (Hap_18) as significant among the selected haplotypes, as the germination percentage and index were both obviously affected by salinity. One Philippines research group identified a total of 28 SNPs (seven of which were in exons) in the gene region of *BADH1* under salt stress screening [14]. After the association test between haplotype groups and salt tolerance traits, we found 15 SNPs of the *BADH1* gene in Indica-specific Hap_18, which was one of the significant haplotypes associated with salt stress. Once all the identified and associated SNPs were verified, we observed that there were only two nonsynonymous substitutions, G/T (exon 11) resulting in amino acid change of ‘Glutamine (Gln) to Lysine (Lys)’ and A/T (exon 4) in the amino acid transition of ‘Asparagine (Asn) to Lysine (Lys)’, and all the remaining were found in different introns. We supposed that these Indica-specific SNPs would be associated with the main functional properties of the *BADH1* gene under salt treatment because of the previous findings that *BADH1* transcript levels were increased in salt-stressed Indica and Japonica nonfragrant rice varieties through its high response to osmotic stress [9].

In conclusion, different directional selections indicated the *BADH1* domestication signature among the classified populations. A greater range of genetic differentiation was observed in the *BADH1* gene region of the cultivated and wild rice group, providing useful genetic information for the upcoming breeding programs of this gene *BADH1*-related varieties development. Haplotyping revealed a list of cultivated haplotypes for a sequence of different mutated variants, of which five major haplotypes showed their associations with salt-conditioned rice seedlings (plant traits) by 27 significant marker positions (SNPs). Hap_18 and Hap_23 represented major groups for Indica rice accessions that covered significant intronic and exonic marker positions (SNPs and Indels) for salt tolerance-related plant traits, which can be future functional *BADH1* alleles in the breeding program of new varieties development.

## 4. Materials and Methods

### 4.1. Plant Materials

A heuristic set of 421 cultivated rice accessions represented by 3 original variety types (landrace, weedy, bred) (Appendix A) previously collected worldwide and generated by the National GenBank of the Rural Development Administration (RDA-GenBank, Republic of Korea) using the Power Core program [27] was selected for whole-genome resequencing [13]. An additional set of 54 wild rice accessions was also shared by the International Rice Research Institute (IRRI) in 2017.

For these 421 Asian cultivated and 54 wild rice accessions, field experiments were conducted in the Departmental Field of the Plant Resources Department, Kongju National University (Yesan Campus) in 2016 and 2017. The landrace, weedy, and bred cultivated rice set included 6 different ecotypes, 279 Temperate Japonica, 26 Tropical Japonica, 102 Indica, 9 Aus, 2 Aromatic, and 3 Admixture accessions (Appendix A). Cultural practices in field management were performed as recommended.

### 4.2. DNA Extraction, Resequencing, and Variant Calling

Fifteen-day-old young samples (green leaves) were taken from all tested plants for DNA extraction by the CTAB (cetyltrimethylammonium bromide) method, and then, genomic DNA was stored in a refrigerator at 4 °C until use [28]. Qualified DNA was used for whole-genome resequencing of the collected rice accessions with an average coverage of approximately 15X on the Illumina HiSeq 2500 Sequencing Systems Platform. The DNA library was generated by using the TruSeq Nano DNA kit, following a specified protocol (part no. 15,041,110 rev. D). The decoded sequences were saved in FastQ file format. VCFtools (variant call format) [29] was used to remove missing values and heterozygotes from raw data saved in FastQ. To compare the output sequences among the accessions, the high-quality reads after removing missing values and heterozygotes were aligned in the International Rice Genome Sequencing Project (IRGSP) 1.0 rice genome sequence. The alignment of the reads was saved in binary alignment map (BAM) format. Duplicate reads aligned in multiple locations were removed using PICARD version 1.88 [30]. Then, variant (SNP/InDel) calling was performed using the Genome Analysis Tool Kit (GATK) tools version 4.0.1.2 [31] to extract the variant regions from the BAM file. The extracted mutations were saved in VCF file format and filtered using VCFtools to remove false-positive SNPs/InDels. The raw sequence data were deposited in the NCBI GeneBank database (accession number: MZ544903-MZ545377). 

### 4.3. Population Structure, Principal Component Analysis (PCA), and Phylogenetic Study

To determine the population structure and existence of subpopulations, we conducted population structure analysis and principal component analysis using 475 rice accessions. We converted annotated variants of *BADH1* into a PLINK file by using VCFtools, and using the PLINK analysis toolset, bed files were recreated, and two additional two files (.bim and .fam) were incorporated by using Python script (structure.py) in the fastStructure package tools [32] within a range of increasing K values from 2 to 7. The admixed patterns of defined populations (population structure) were implemented using average Q-values by the POPHELPER [33] analytical tool in the R program. To plot the similarity or differences among genetic variations in the identified subpopulations, principal component analysis (PCA) was performed in the R program. A list of principal components (PCs) referring to variants was generated from TASSEL 5 [34], and the relatedness among the groups was plotted in 3D scatterplots. A phylogenetic analysis was conducted in MEGAX [35] by the neighbor-joining method, and a tree was drawn in FigTree version 1.4.3 (http://tree.bio.ed.ac.uk/software/figtree, accessed on 18 January 2021).

### 4.4. Nucleotide Diversity, Tajima’s D, and Fixation Index (F_ST_)

To determine the genetic diversity, differentiation, and variation differences, we calculated the respective values for the nucleotide diversity (π), Tajima’s D, and the fixation index (*F_ST_*). Using VCFtools, variant files were picked within the gene region of *BADHI* for the classified representative types/subpopulations to be compared. The sliding window sizes used for nucleotide diversity (π) and Tajima’s D test were each 10 kb, and the values were compared in multiple ways. *F_ST_* values were also calculated to determine the genetic differentiation between and among the identified groups or subpopulations of 475 rice accessions.

### 4.5. Haplotype Diversity Analysis

We conducted whole-genome haplotype diversity analysis on *BADHI* using a variant annotation file to identify the association between *BADH1* and salt tolerance at the haplotype level. We divided our sample group into two separate groups, cultivated rice and wild rice, and performed haplotyping individually. Haplotyping of cultivated rice was first conducted and then compared to the results for wild rice with a minor allele frequency (maf) filter of <0.03 maf. The sequence data from both groups were aligned in MEGAX together with the reference sequence adapted from RAP-DB (https://rapdb.dna.affrc.go.jp, accessed on 30 January 2021), and a haplotype list was generated by DnaSP version 6.0 [36] Using the filtered and aligned genome sequences, a TCS network [37] was constructed in PopART [38].

### 4.6. Screening of Salt Tolerance Phenotypes

First, a total of 120 seeds of each rice accession were washed in water, surface-sterilized in 1% sodium hypochlorite solution for 20 min, and then rinsed three times with deionized distilled water. Thirty seeds were taken from each rice accession and placed in 9 cm diameter Petri dishes supplemented with 200 mM NaCl solution for salt stress and two layers of filter paper underneath. Then, Petri dishes were stored in incubators at 30 °C with 40% relative humidity for 10 days. Every 2 days, the NaCl solution in Petri dishes was renewed to maintain the concentration, and the germination conditions were checked every day. Filter papers were replaced as necessary. Once the plumule emergence was 2 mm long, we started to measure it as the germination index (GI). After 10 days, we measured the root length (RL) and shoot length (SL) of the seedlings. Then, the total dry weight (TDW) of roots and shoots was also measured after drying at 80 °C for 24 h. The data of this treatment were collected for three replicates, and 0 mM NaCl was used as a control. The number of germinated seeds was counted every day after treatment for up to 10 days, and the germination percentage (GP) was calculated. Germination energy (GE) was observed and recorded daily for 4 days, and the values were calculated. The formulae we used are summarized in Table 3.

### 4.7. Statistical Analysis

The recorded phenotypic data were first calculated in Microsoft Excel (2010) and statistically analyzed in SPSS version 20.0 using Pearson correlation coefficients. Haplotypic and phenotypic data files were prepared and imported to TASSEL 5.0 [33] for the association test. The general linear model (GLM), containing the SNP tested as a fixed effect, was applied to test the association between phenotypic variation and haplotypes. The association between phenotype and genotype was obtained by using Scheffe’s test at the significance level (*p*-value < 0.05).

## Figures and Tables

**Figure 1 ijms-22-07578-f001:**
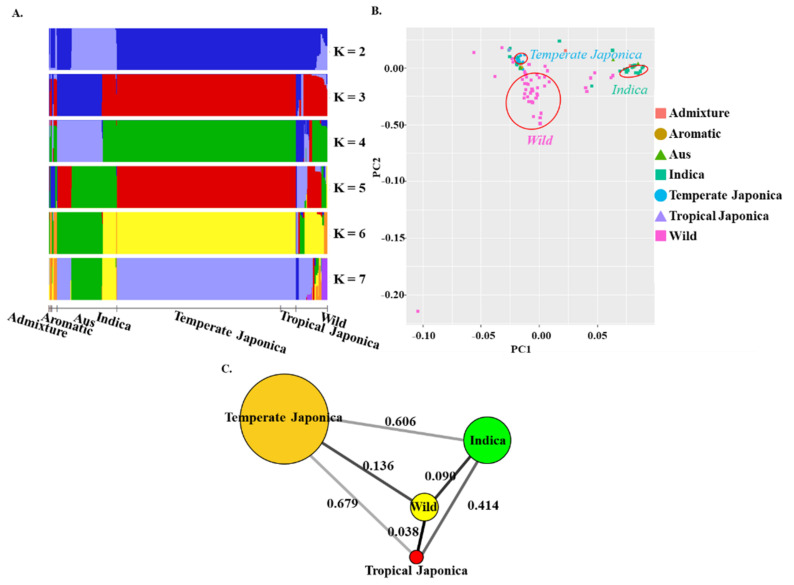
Population structure and diversity in *BADH1* (*Os04g0464200*). (**A**) The population structure of all subgroups based on the *BADH1* region using fastStructure to estimate individual ancestry and admixture proportions assuming K populations. With increasing K (number of populations) values from 1 to 10 with 10 iterations each, we analyzed the population structure for each K value (from K = 2 to 7). (**B**) Principal component analysis (PCA) analysis of seven classified ecotypes. Indica and Japonica clustered into clearly defined groups, and other subgroups were located around the two groups. PCA was performed using TASSEL and plotted with ggplot2. (**C**) *F_ST_* values between the four classified ecotypes, indicating circle size by the number of accessions. The dark line between each pair represents their genetic distance.

**Figure 2 ijms-22-07578-f002:**
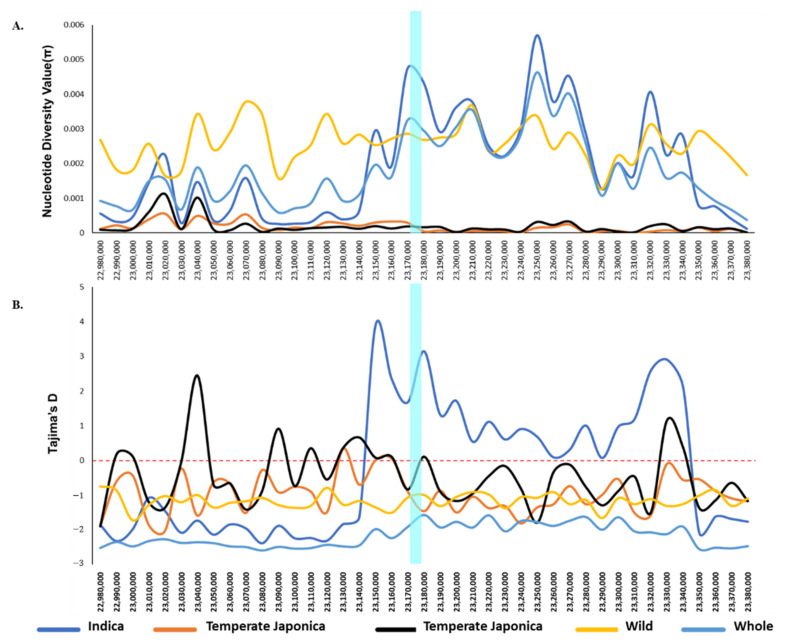
Nucleotide diversity (π) values and Tajima’s D values of *BADH1* (*Os04g0464200*) in 475 rice accessions. (**A**) Nucleotide diversity (π) indicating different pi (π) values at segregating sites of different classified groups determined with a 10 kb sliding window within the *BADH1* region (**B**) Tajima’s D values indicating different variations among the classified groups identified by a 10 kb sliding window within the *BADH1* region. Cyan color indicates the *BADH1* gene region; each colored line represents different rice groups in terms of ecotypes.

**Figure 3 ijms-22-07578-f003:**
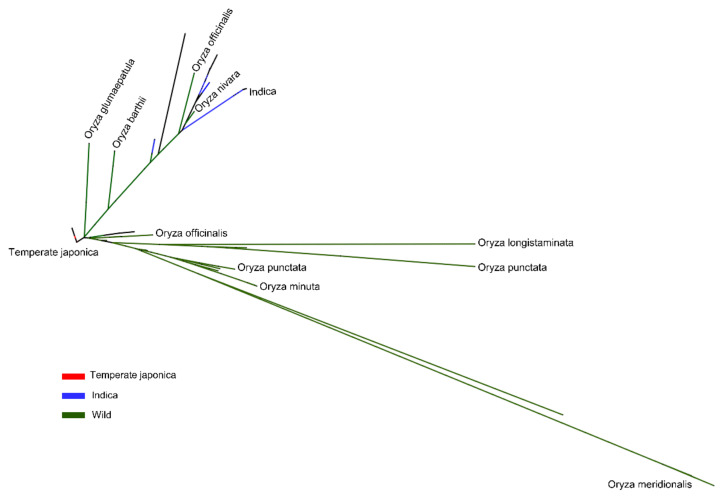
Phylogenetic tree of *BADH1* gene in 475 rice accessions. The classified groups of cultivated rice were considered only in terms of major ecotypes, namely, Indica, Temperate Japonica, and Tropical Japonica, and their phylogenic relationship with wild rice. The tree was constructed for genetically different sequences of the *BADH1* gene, using MEGAX software. The reliability of the neighbor-joining phylogeny output was estimated using bootstrap analysis with 1000 permutations.

**Figure 4 ijms-22-07578-f004:**
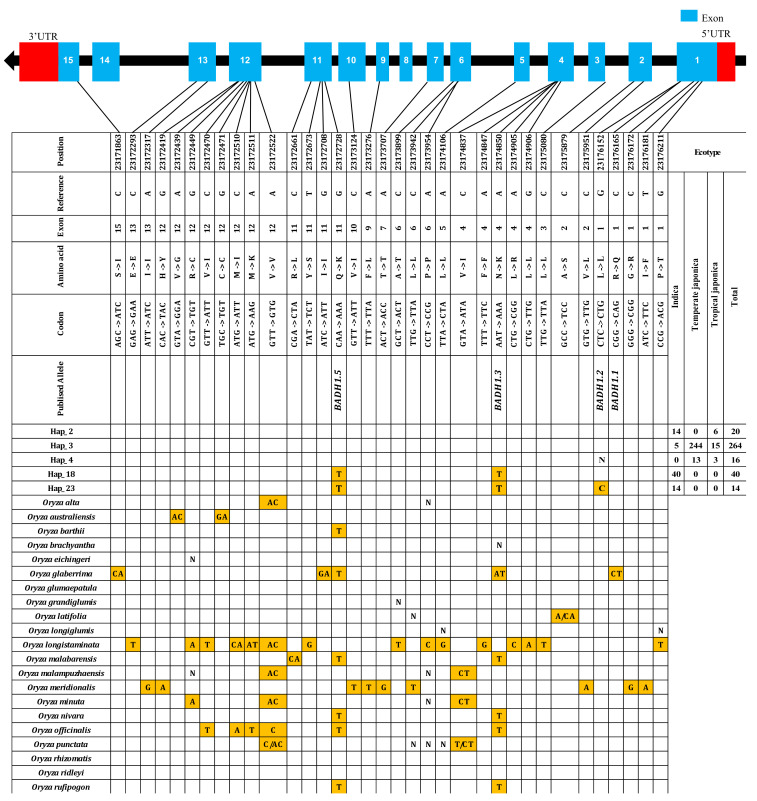
Gene structure and haplotype analysis of *BADH1* (*Os04g0464200*) in gene-coding regions of selected cultivated rice and all wild species. The five cultivated haplotypes representing the highest number of rice accessions were selected, and their genetic association was investigated by SNPs or InDels in wild rice. Here, we used one wild rice accession as a haplotype. Orange-colored SNPs indicate all identified SNPs observed as nonsynonymous and synonymous substitutions. Blank cells refer to the same nucleotide mentioned in reference. “N” indicates the positions of unknown nucleotides. The two-digit number indicated under ecotype (Indica, Temperate Japonica, and Tropical Japonica) refers to the number of rice accessions identified in each haplotype.

**Figure 5 ijms-22-07578-f005:**
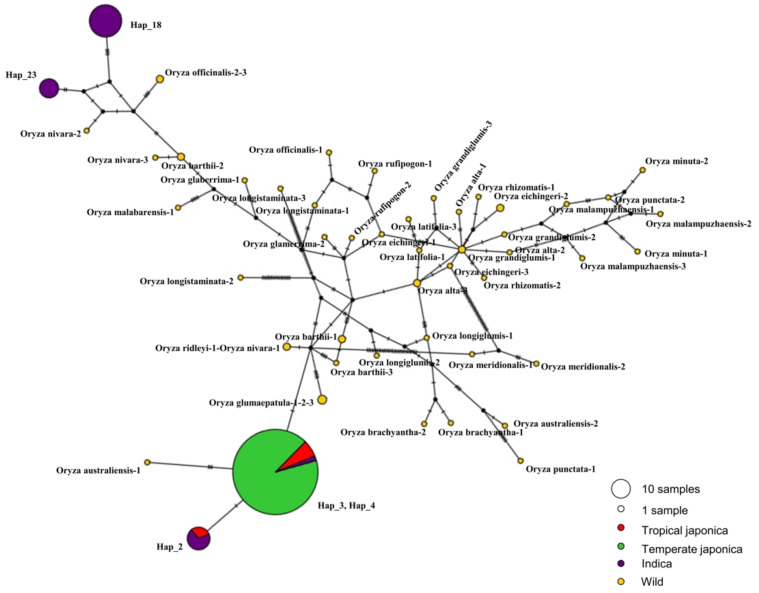
Haplotype network visualizing possible genotypic relationships of major cultivated haplogroups compared to wild rice haplotypes within the *BADH1* region. The size of each circle is proportional to the accession numbers encompassed, and different colors indicate its ecotype. The median vectors indicated by the black circular dot is a hypothetical sequence used to connect the existing similar sequences.

**Figure 6 ijms-22-07578-f006:**
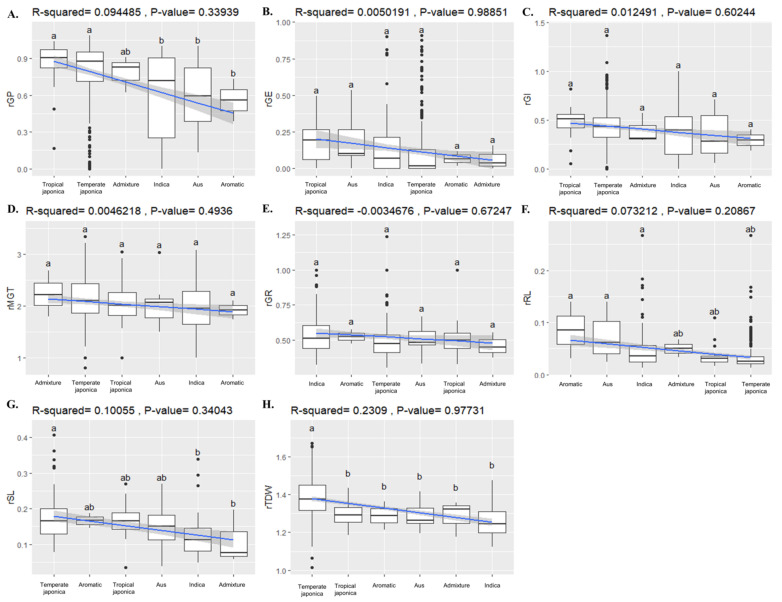
Test/Control ratio of eight major plant parameters distribution group by ecotypes. (**A**–**H**) Relative variations of salt tolerance-related traits among the classified ecotypes, indicating significant level by statistical test. The order of classified ecotypes was sorted (from left to right) by their relative/statistical mean value (from highest to lowest) of each trait. Each parameter was drawn in a boxplot using 417 rice accession for which phenotype data were surveyed. For each parameter, the analysis was performed at a significant level of *p*-value < 0.05, and the values were compared among the classified rice ecotypes using Scheffe’s test and indicated on the boxplot of each ecotype. Abbreviations: rGP, relative germination percentage; rGE, relative germination energy; rGI, relative germination index; rGMT, relative germination mean time; rGR, relative germination rate; rRL, relative root length; rSL, relativce shoot length; rTDW, relative total dry wight.

**Figure 7 ijms-22-07578-f007:**
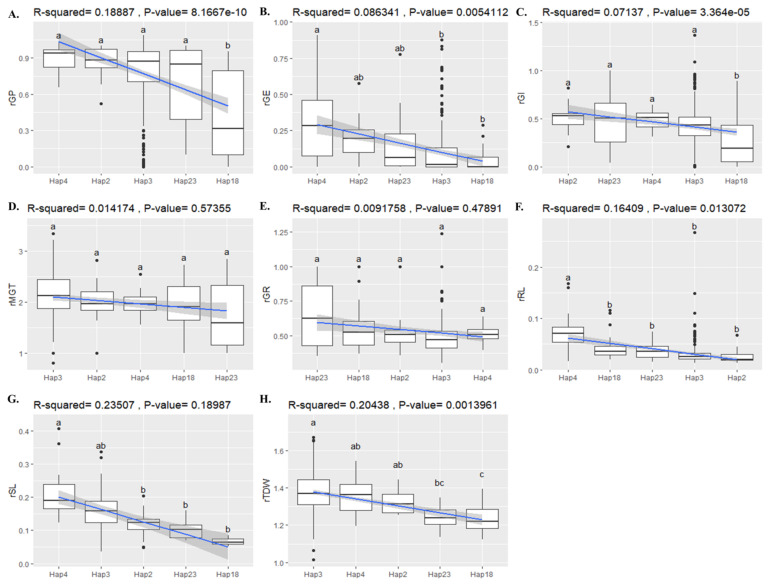
Association of selected *BADH1* haplotypes and eight plant parameters under salt treatment. (**A**–**H**) Relative variations of salt tolerance-related traits among the selective predominant haplotypes, indicating significant level by statistical test. The order of these predominant haplotypes was sorted (from left to right) by their relative/statistical mean value (from highest to lowest) of each trait. Among the previously identified 39 haplotypes of cultivated rice, only five predominant haplotypes were selected for comparison in association to each salt tolerant parameters due to the rice accession number to which they belonged. For association level to each plant parameter, the selected haplotypes were compared by using Scheffe’s test at a significant level (*p*-value < 0.05). Abbreviations: rGP, relative germination percentage; rGE, relative germination energy; rGI, relative germination index; rGMT, relative germination mean time; rGR, relative germination rate; rRL, relative root length; rSL, relativce shoot length; rTDW, relative total dry wight.

**Table 1 ijms-22-07578-t001:** Summary of the total variations (SNPs, InDels, and SV) detected in the *BADH1* gene region of 421 cultivated and 54 wild rice accessions. Te: Temperate; Tr: Tropical.

Ecotype	SNP	Insertion	Deletion	Structure Variation	Total Accessions
Te_Japonica	0	1	0	0	279
Tr_Japonica	1	1	0	0	26
Indica	23	2	1	0	102
Aus	23	2	0	0	9
Aromatic	0	1	0	0	2
Admixture	15	2	0	0	3
Wild	105	38	36	2	54

**Table 2 ijms-22-07578-t002:** Descriptive statistics for the mean value of the traits in the control and salt-treated (200 mM) rice accessions. GP: Germination Percentage; GE: Germination Energy; GI: Germination Index; GMT: Mean Germination Time; GR: Germination Rate; SL: Shoot Length; RL: Root Length; TDW: Total Dry Weight; SD: Standard Deviation; IQR: Interquartile Range.

Trait	Salinity Level (mM)	Mean ± SD	Range	Median	IQR
GP	200	21.57 ± 7.94	0–30	24.33	29.517
	0	28.47 ± 2.72	0–30	29	29.585
GE	200	3.17 ± 5.37	0–27	0.33	28.0218
	0	22.93 ± 10.19	0–30	28	48.131
GI	200	3.55 ± 1.84	0–9.24	3.59	7.5618
	0	8.49 ± 2.28	0–14.83	9.27	13.21
MGT	200	7.31 ± 1.57	3.34–10	7.24	6.7405
	0	3.87 ± 1.83	2–10	3.14	6.1583
GR	200	0.14 ± 0.03	0.1–0.3	0.14	0.2735
	0	0.29 ± 0.07	0.1–0.5	0.32	0.4516
SL	200	0.36 ± 0.27	0.19–2.34	0.25	2.1924
	0	10.13 ± 2.93	0.98–19.43	9.89	15.911
RL	200	0.87 ± 0.51	0.2–8.11	0.81	6.7159
	0	5.92 ± 1.31	1.86–11.01	5.67	7.854
TDW	200	0.68 ± 0.08	0.36–1.05	0.68	0.9325
	0	0.51 ± 0.07	0.26–0.78	0.5	0.695

**Table 3 ijms-22-07578-t003:** Plant parameters (phenotypes) and formulae for their calculation.

Parameters (Phenotypes)	Formula
Germination percentage (GP)	(number of germinated seeds/total number of seeds) × 100
Germination energy (GE)	(number of germinated seeds on day 4/total number of seeds) × 100
Germination index (GI)	Σ (N_d_/d)
Mean germination time (MGT)	Σ (d × n)/Σ n_d_ n_d_: the number of germinated seeds on each day d: number of days after the start of the experiment
Germination rate (GR)	Σ N/Σ (N × d) N: the number of seeds that germinated on day d d: the days during the experiment
Root length (RL)	Length of root after 10 days
Shoot length (SL)	Length of shoot after 10 days
Total dry weight TDW)	Total dry weight of shoot and root (80 °C for 24 h)
Relative GP	GP in condition/GP in control
Relative GE	GE in condition/GE in control
Relative GI	GI in condition/GI in control
Relative MGT	MGT in condition/MGT in control
Relative GR	GR in condition/GR in control
Relative RL	RL in condition/RL in control
Relative SL	SL in condition/SL in control
Relative TDW	TDW in condition/TDW in control

## Data Availability

Data is contained within the article or Appendix A.

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
