# Peer review of "Haplotype Analysis of BADH1 by Next-Generation Sequencing Reveals Association with Salt Tolerance in Rice during Domestication"

_ijms, 2021, doi:10.3390/ijms22147578_

Round 1

Reviewer 1 Report

Manuscript is well written and interesting. It is a very consistent work from an experimental point of view, full of well-integrated multidisciplinary approaches. The sampling on which the investigations were carried out is numerous and consistent and therefore the results can be considered significant. The introduction is well written and introduces the reader well to the problem. The materials and methods paragraph is clear and the experimental part well described including the computer analyzes performed. The results are clear, well described and the subdivision into paragraphs makes the reading understandable. The discussion paragraph, on the other hand, should be revised. I had difficulty in understanding what were the conclusive sentences with which the authors argued the results obtained in relation to the problem analyzed. Above all, the relationship between salt tolerance and the phenomenon of domestication should be better argued. I would probably suggest better introducing the rice domestication pathway over the years and subsequently arguing for the potential relationship they observe for salt tolerance.

Author Response

Response to Reviewer 1 Comments

Point 1: Manuscript is well written and interesting. It is a very consistent work from an experimental point of view, full of well-integrated multidisciplinary approaches. The sampling on which the investigations were carried out is numerous and consistent and therefore the results can be considered significant. The introduction is well written and introduces the reader well to the problem. The materials and methods paragraph is clear and the experimental part well described including the computer analyzes performed. The results are clear, well described and the subdivision into paragraphs makes the reading understandable. The discussion paragraph, on the other hand, should be revised. I had difficulty in understanding what were the conclusive sentences with which the authors argued the results obtained in relation to the problem analyzed. Above all, the relationship between salt tolerance and the phenomenon of domestication should be better argued. I would probably suggest better introducing the rice domestication pathway over the years and subsequently arguing for the potential relationship they observe for salt tolerance.

Response 1: Thanks to the reviewer for the suggestions given. We have updated the contents in the discussion as advised (lines 382-403). We have also revised the unclear descriptions for the concluding sentences in the last paragraph of the discussion section (lines 443-450).  Regarding domestication, we first added an introduction for the domestication of rice (lines 382-392). Many genes are related to salt tolerance and some of them might be affected by domestication. But one of our works is the first report about BADH1’s domestication. Nevertheless, we compared our result with the arguments regarding domestication about the other genes related to salt tolerance including the potential genes presented in the previous studies. So we have added the content dealing with the arguments that the genes related to salt tolerance did not have the same domestication pathway overall (lines 382-403).

Note: Please select “simple markup” if you use “track changes” function in manuscript so that line numbers will be the same to that we indicated.

Reviewer 2 Report

The manuscript entitled “Haplotype Analysis of BADH1 by Whole Genome Resequencing Reveals Association with Salt Tolerance in Rice During Domestication” deals with characterization of haplotypes in BADH1 gene in rice and their associations with salt tolerance. The manuscript presents solid data and methodological approaches seems to be adequate. However, I have several comments to the concept, and data presentation.

Major points

1. The authors used whole-genome sequencing of 475 rice accessions to investigate BADH1 haplotypes, as this gene is suspected to be involved in salt tolerance. This approach ignores the most genomic data and focuses on BADH1 gene only. Why to focus on the ony selected gene, if the whole genome data is available? I do not understand, why the authors did not use for example GWAS analysis - at least for the selection of promising genomic regions. Generally, genomic context is not considered in some analyses. In the chapter 2.3, nucleotide diversity was analysed for predefined subpopulations and it shows that this diversity is much more higher in BADH1 gene region in indica compared to japonicas group. However, only relatively tight region of the chromosome 4 is considered (Fig.2). Thus it is not clear if this diversity increment is linked specifically to this region, or if it is a characteristic of the whole genome.

2. The reasons for particular analyses are unclear and the results are not commented in terms of tested hypotheses. It is the case of the population structure and phylogenetic analysis in chapters 2.2 or 2.4.

Similarly, there is a question, why the authors try to correlate plant characteristics, that are linked with salt tolerance. Is so surprising that they correlate, if all these parameters had the same trend of responses to salt treatment? And if so, what does it mean?

3. Conclusions are not presented unambiguously. For example, the authors identified Hap_18 as haplotype significantly affected by salt treatment for parameters RGP a RGI. However Hap_18 shows very high variability in this parameter and according to the tables S4 and S7 detected SNPs include also synonymous variants although they associate with measured parameters, which is surprising and an explanation should be suggested.

Minor points

4. Right part of the figure 4 (with Hap frequencies in particular ecotypes) could be presented separately for better overview.

5. Abbreviations for relative parameters are not explained in the text and maibe rGP (?) would be more understandable then RGP.

Author Response

Response to Reviewer 2 Comments

The manuscript entitled “Haplotype Analysis of BADH1 by Whole Genome Resequencing Reveals Association with Salt Tolerance in Rice During Domestication” deals with characterization of haplotypes in BADH1 gene in rice and their associations with salt tolerance. The manuscript presents solid data and methodological approaches seems to be adequate. However, I have several comments to the concept, and data presentation.

Major points

Point 1: The authors used whole-genome sequencing of 475 rice accessions to investigate BADH1 haplotypes, as this gene is suspected to be involved in salt tolerance. This approach ignores the most genomic data and focuses on BADH1 gene only. Why to focus on the only selected gene, if the whole genome data is available? I do not understand, why the authors did not use for example GWAS analysis - at least for the selection of promising genomic regions. Generally, genomic context is not considered in some analyses. In the chapter 2.3, nucleotide diversity was analyzed for predefined subpopulations and it shows that this diversity is much more higher in BADH1 gene region in indica compared to japonicas group. However, only relatively tight region of the chromosome 4 is considered (Fig.2). Thus, it is not clear if this diversity increment is linked specifically to this region, or if it is a characteristic of the whole genome.

Response 1: We thank the reviewer for the professional suggestions made and clear-cut points. We have also understood what the reviewer intended in providing advice based on our data base. We changed a part of title, “Whole Genome Resequencing” to “Next-Generation Sequencing” so that it will not be misleading. We have studied the fragrance of rice using Next-Generation Sequencing (NGS) data. While studying BADH2, the driving gene for fragrance, we found out that BADH1 is evolutionary and related to BADH2 despite their functions being totally different (Qiang He et al., 2015). Now, focusing on BADH1, a familiar gene for salt tolerance, we defined the high-quality haplotypes of BADH1 consisting of a couple of variant sites at the nucleotide level and figured out the correlations between salt tolerance and the haplotypes of BADH1. So, we have updated the SNPs and indel variant information using variant calling from our NGS database of the rice core collection consisting of 475 accessions. We have used the variant information to obtain high quality genotypes of SNPs and indels in the BADH1 region. For this purpose, the results for all the analyses were particularly considered only for that specific gene region (exactly including the upstream and downstream sequence from BADH1 for the analyses), using classified subpopulations.

Point 2: The reasons for particular analyses are unclear and the results are not commented in terms of tested hypotheses. It is the case of the population structure and phylogenetic analysis in chapters 2.2 or 2.4.

Similarly, there is a question, why the authors try to correlate plant characteristics, that are linked with salt tolerance. Is so surprising that they correlate, if all these parameters had the same trend of responses to salt treatment? And if so, what does it mean?

Response 2:  We thank the reviewer for the systematic and specific consideration of our work. The reason why we conducted analyses on BADH1 in terms of population structure and phylogeny was to figure out and understand the BADH1 gene’s specific populational distribution, phylogenetic origin and relationships which are the representations of the phylogenic and evolutionary trace that has made for the current diversity of BADH1, and the analyses linked to the haplotype diversity. For the test hypotheses, we added the contents of the tested hypotheses for ANOVA (Supplementary Table S8A and B) and Scheffe’s test (Figure 6 and 7).

The quantitative degree of salt tolerance cannot be measured easily. A quantitatively measurable character such as shoot length and root length represents an aspect of salt tolerance. However, it does not mean salt tolerance by itself. In other words, it cannot cover the entire representation of salt tolerance. Therefore, by using multiple measurable characters salt tolerance can be estimated quantitatively. And the growth of the germination stage of rice can be evaluated by the combination of all those parameters. In this sense, the salt tolerant plants’ germination rate will be higher under salt stress than that of the normal plants. However, it can be expected that the degree of representation for salt tolerance differs between the parameters, and in order to figure out which is most responsive to salt treatment, we measured the correlations between those parameters.

Point 3: Conclusions are not presented unambiguously. For example, the authors identified Hap_18 as haplotype significantly affected by salt treatment for parameters RGP a RGI. However, Hap_18 shows very high variability in this parameter and according to the tables S4 and S7 detected SNPs include also synonymous variants although they associate with measured parameters, which is surprising and an explanation should be suggested.

Response 3: We highly appreciate the reviewer for pointing out very reasonable aspects of our result presentation of the haplotype analysis. We revised the concluding sentences based on what we found in a series of analyses to avoid any ambiguous conclusions in line with the reviewer’s comment (lines 443-450). We gave weight to the number of accessions for the association test.  We selected 5 haplotypes (Hap_2, 3, 4, 18 and 23) considering the number of accessions to be more than five rice accessions. And Hap_18 showed a significant association with tested plant parameters, GP and GI, indicating two non-synonymous SNPs (G/T in exon 11 and A/T in exon 4) (Supplementary Table S4).

Note: Please select “simple markup” if you use “track changes” function in manuscript so that line numbers will be the same to that we indicated.

Minor points

Point 4: Right part of the figure 4 (with Hap frequencies in particular ecotypes) could be presented separately for better overview.

Response 4: Thank you for pointing out that this information should be provided separately. We have updated the contents in line with the reviewer’s comment (lines 206-214).

Point 5: Abbreviations for relative parameters are not explained in the text and maibe rGP (?) would be more understandable then RGP.

Response 5: Thank you for this suggestion. As suggested by the reviewer, we have changed the texts in the abbreviations (lines 311-323, including all the others, figures and tables).

Round 2

Reviewer 2 Report

The manuscript was improved, but according to me, some important information are still missing.

Major points

- I have to say that I am a little bit confused from the author‘s response. It seemed from the first manuscript version, that the authors made Next-Generation Sequencing of studied rice accessions, but now in the response, they claimed that only SNPs and indel variant information from NGS database was updated using variant calling. However, variant calling is only bioinformatic tool. In the chapter 4.2 of revised manuscript version, it is declared, that whole-genome sequencing was done. Was NGS performed, or not? It must be declared absolutely precisely in the text of the manuscript (including appropriate references and/or accession links), not only in the response. If NGS was performed newly, the authors must add a procedure of NGS library preparation.

- Additionally, the authors did not comment the reasons, why only relatively tight region of the chromosome 4 was considered. For example, Garris et al., 2005 (PMC1449546) showed that The temperate japonica and aromatic groups had lower diversity with 88% polymorphic loci. It shows that these groups has generally reduced nucleotide diversity (at the whole genome level), not only diversity in BADH1, which should be discussed.

- I can understand, why salt tolerance-related parameters could be correlated in a “plant physiology study”, but explanation offered by the authors does not make sense in this study. Why are correlations necessary for disclosure of the most responsive parameter to salt treatment (if it is important, it must be commented in the manuscript)? The reason, why these correlations were calculated, remains unclear. The authors should explain it clearly, or (better) remove it from the manuscript.

- The sixth paragraph of discussion seems to be partially misleading.

- - l. 440 - the sentence “One Philippines…” is not understandable … what do the words “resulting in ...” mean? Because in the paper, there is no BADH1 expression difference between studied groups (SNPs) reported.

- - The conclusionThese indica-specific SNPs may be associated with the main functional properties of the BADH1 gene under salt treatmentis very controversial. If non-synonymous mutations can change main functional properties of the BADH1 gene under salt treatment, then we could expect, that all carriers of these variants will have the same properties (especially if all other SNPs are intronic). However, if I understand it well, these SNPs are present also in Hap_23 and according to the Fig. 7A and 7C, these properties are different. I would suspect from these indications, that these SNPs can not be responsible for salt-sensitivity of Hap_18. If the most of other variants are intronic variant, we can imagine that they influence “the main functional properties” only by changing expression level. In this case, the authors must analyze the expression level of BADH1 in these accessions. It could provide unambiguous proof, if changes in BADH1 expression/function can mediate salt-sensitivity of Hap-18, or if this behavior is mediated by other factor/gene, which is for example genetically linked with Hap-18 (?).

Author Response

Response to Reviewer 2 Comments

Point 1: I have to say that I am a little bit confused from the author‘s response. It seemed from the first manuscript version, that the authors made Next-Generation Sequencing of studied rice accessions, but now in the response, they claimed that only SNPs and indel variant information from NGS database was updated using variant calling. However, variant calling is only bioinformatic tool. In the chapter 4.2 of revised manuscript version, it is declared, that whole-genome sequencing was done. Was NGS performed, or not? It must be declared absolutely precisely in the text of the manuscript (including appropriate references and/or accession links), not only in the response. If NGS was performed newly, the authors must add a procedure of NGS library preparation.

Response 1: We thank the reviewer for valuable suggestions made and clear-cut professionally. We also agreed to understand on what reviewer intended to advise based on our data base. We apologize for using confused word in chapter 4.2 and appreciated again on focusing word. In fact, this was not “whole-genome resequencing”, not “sequencing” (line 476) and we have corrected it in the revised version. As of reviewer’s comment, we also updated the contents in line with NGS library preparation (lines 477-478).

As an additional clarification about the data usage, we have submitted the raw sequencing data (BADH1 gene FASTA files for 475 rice accessions) we have dissected in the manuscript at the GenBank (NCBI Resource Coordinators, 2016) Sequence Read Archive (SRA). But it is regrettable to inform that our submitted data are not yet ready to get access due to its under processing status. So, we also attached this captured image (see uploaded file) of “the GeneBank Submissions” to reflect on the reviewer’s consideration. We added the information about the publicity of the data we used in this manuscript (line 489) and then, we will update the confirmed accession link when it is ready.

Point 2: Additionally, the authors did not comment the reasons, why only relatively tight region of the chromosome 4 was considered. For example, Garris et al., 2005 (PMC1449546) showed that The temperate japonica and aromatic groups had lower diversity with 88% polymorphic loci. It shows that these groups has generally reduced nucleotide diversity (at the whole genome level), not only diversity in BADH1, which should be discussed.

Response 2: We thank the reviewer for clear pointing out. As reported earlier, we wanted to identity BADH1 from using variant calling from our NGS database of the rice core collection consisting of 475 accessions. Then, we picked up only BADH1 gene region (chromosome 4) for data requirements and we used it in a series of analyses for the purposes of investigating domestication signals of that gene, BADH1. For such purposes, the results for all analyses were particularly considered only in that specific gene region (chromosome 4 for BADH1), using classified subpopulations.

Point 3: I can understand, why salt tolerance-related parameters could be correlated in a “plant physiology study”, but explanation offered by the authors does not make sense in this study. Why are correlations necessary for disclosure of the most responsive parameter to salt treatment (if it is important, it must be commented in the manuscript)? The reason, why these correlations were calculated, remains unclear. The authors should explain it clearly, or (better) remove it from the manuscript.

Response 3: We apologize for our previous unclear explanations about the correlation analysis but thanks to the reviewer for the reasonable suggestion with this. We have moved it into supplementary files, instead we presented the resulted mean values of tested traits under salt and control conditions (lines 256-275).

Point 4: The sixth paragraph of discussion seems to be partially misleading.

Response 4: Thank you for reviewer’s suggestion with this paragraph. Please check that we have updated this paragraph with reasonable revised contents (line 447-457)

Point 4-1: l. 440 - the sentence “One Philippines…” is not understandable … what do the words “resulting in ...” mean? Because in the paper, there is no BADH1 expression difference between studied groups (SNPs) reported.

Response 4-1: Thank you for clarifying that reference sentence. We also apologize for wrong number of SNPs we cited. This cited paper identified 28 SNPs in the polymorphism of BADH1 gene regions which in turn to express higher transcript level under saline condition, compared with normal condition without salt (line 434-436). Our finding, one of associated haplotypes, Hap_18, indicated a total of 15 SNPs which could be expected to be significant in future gene expression analysis, but this time, we did focus only on association analysis.

Point 4-2: The conclusion “These indica-specific SNPs may be associated with the main functional properties of the BADH1 gene under salt treatment” is very controversial. If non-synonymous mutations can change main functional properties of the BADH1 gene under salt treatment, then we could expect, that all carriers of these variants will have the same properties (especially if all other SNPs are intronic). However, if I understand it well, these SNPs are present also in Hap_23 and according to the Fig. 7A and 7C, these properties are different. I would suspect from these indications, that these SNPs can not be responsible for salt-sensitivity of Hap_18. If the most of other variants are intronic variant, we can imagine that they influence “the main functional properties” only by changing expression level. In this case, the authors must analyze the expression level of BADH1 in these accessions. It could provide unambiguous proof, if changes in BADH1 expression/function can mediate salt-sensitivity of Hap-18, or if this behavior is mediated by other factor/gene, which is for example genetically linked with Hap-18 (?).

Response 4-2: We agreed with the opinions you pointed out and appreciated too. In this time analysis, we investigated our data for the association between the classified haplotypes and tested plant traits. Based on their significances in association levels, we identified the representative variants and evaluated for their possibilities in future analyses. As reviewer suggested, we imagined checking the expression levels of each marker to be confirmed for their functional properties. Unfortunately, the main objective of this analysis was to observe the association between the haplotypic and phenotypic data. So. We may continue to do these analyses in future research study investigating the details of identified alleles.

Note: Please select “simple markup” option if you use “track changes” function in manuscript so that line numbers will be the same to that we indicated.
